# Multifactorial Predictors of Late Epileptic Seizures Related to Stroke: Evaluation of the Current Possibilities of Stratification Based on Existing Prognostic Models—A Comprehensive Review

**DOI:** 10.3390/ijerph18031079

**Published:** 2021-01-26

**Authors:** Adam Wiśniewski, Dalius Jatužis

**Affiliations:** 1Department of Neurology, Faculty of Medicine, Nicolaus Copernicus University in Toruń, Collegium Medicum in Bydgoszcz, 85-094 Bydgoszcz, Poland; 2Center of Neurology, Vilnius University, LT-08661 Vilnius, Lithuania; dalius.jatuzis@mf.vu.lt

**Keywords:** epilepsy, stroke, late seizures, cerebrovascular disease, prognosis, predictors, prevention, risk score

## Abstract

*Background*: Epilepsy associated with strokes is a significant clinical and public health problem and has a negative impact on prognosis and clinical outcome. A late epileptic seizure occurring seven days after stroke is actually equated with poststroke epilepsy due to the high risk of recurrence. Predictive models evaluated in the acute phase of stroke would allow for the stratification and early selection of patients at higher risk of developing late seizures. *Methods*: The most relevant papers in this field were reviewed to establish multifactorial predictors of late seizures and attempt to standardize and unify them into a common prognostic model. *Results*: Clinical and radiological factors have become the most valuable and reproducible predictors in many reports, while data on electroencephalographic, genetic, and blood biomarkers were limited. The existing prognostic models, CAVE and SeLECT, based on relevant, readily available, and routinely assessed predictors, should be validated and improved in multicenter studies for widespread use in stroke units. *Conclusions*: Due to contradictory reports, a common and reliable model covering all factors is currently not available. Further research might refine forecasting models by incorporating advanced radiological neuroimaging or quantitative electroencephalographic analysis.

## 1. Introduction

Epilepsy and cerebrovascular diseases are the most common neurological disorders worldwide and constitute a significant public health problem. In the elderly, there is an increasing tendency for both these pathologies to coexist [1]. It is believed that cerebrovascular diseases, especially strokes, may account for about 10% of all epilepsy cases, while in people over 60 it is estimated that they may be the underlying cause of more than half of the cases of newly diagnosed epilepsy [2]. The most recognized classification regarding the division of epilepsy associated with stroke was proposed by the International League against Epilepsy. They subdivided seizures into early seizures, which occur up to seven days after the onset of the stroke symptoms, and late seizures, which may develop more than seven days after stroke onset [3]. Both types of seizures, though related to the same disease, show significant differences in pathological mechanisms, prognosis, and even treatment, which means that they should be considered separately [4]. Early seizures are treated as symptomatic, provoked, and resulting from the course of the acute phase of stroke, and are caused by biochemical changes (ion channel dysfunction, blood-brain barrier disruption, and neurotransmitter release) in damaged brain tissues in response to acute hypoxia [5,6]. Late seizures are associated with the reconstruction and organization of necrotic brain tissue (neurodegeneration, chronic inflammation, gliosis, and angiogenesis), which result in the formation of a focus predisposing to epileptic discharge [7]. A single late seizure, with more than 60% risk of recurrence within the next 10 years, is actually equated with poststroke epilepsy, in accordance with the fulfillment of a new practical clinical definition of epilepsy [3]. Compared to other types of epilepsy, stroke-related seizures are relatively amenable to pharmacological therapy and are characterized by a low rate of drug resistance and a high rate of treatment efficacy. However, a common feature is that both early and late seizures are associated with a worse prognosis and clinical outcome, greater disability, and greater mortality [8,9,10,11].

Such a high incidence and negative impact of epilepsy associated with stroke makes it a significant clinical problem of considerable interest to many researchers. Particular attention is paid not only to the pathomechanism, classification, and treatment methods, but most of all to the predictors that enable the identification of a group at high risk of developing epilepsy. Predictive models and risk scoring at the acute stage of stroke would enable stratification of patients and early selection of patients in whom it could be necessary to implement management preventing the transformation of an ischemic or hemorrhagic focus into an epileptogenic focus. Despite many studies in this field on multifactorial predictors of stroke-related epilepsy, there has still been no attempt to standardize and consolidate them into a common and unified prognostic model. The development of such a model, and associated guidelines for management, would contribute to more effective prevention of the negative effects and complications of this phenomenon.

The aim of this study was to establish the current predictors of late seizures related to stroke and discuss the possibility of their standardization and unification. Moreover, particular attention was paid to the multifactorial background of these predictors to emphasize the complexity of the phenomenon discussed.

## 2. Materials and Methods

This review included relevant papers on late seizures related to stroke, especially regarding the prognosis, predictors, and prevention. Bibliographic databases were searched by two independent researchers. English-language reports of all types from the last 20 years, in particular clinical trials, original research, observational studies, cohort studies, cross-sectional studies, reviews, and meta-analyses were included. There are many valuable reports on this topic from 20 years ago and earlier, but up-to-date reports are needed. The subsequent stages of article selection are shown in Figure 1.

Four databases were searched on the basis of the title and abstract. We excluded records after duplicates, papers that do not focus on stroke and predictors of late epileptic seizures, analyze only early epileptic seizures, or perform no follow-up. Articles that met the preliminary criteria were fully analyzed in terms of their eligibility. At this stage, after a thorough analysis of full-text articles, we decided to exclude papers based on a small study population <300 and not providing any significant findings in this field. Ultimately, only articles that met all the requirements were finally included.

## 3. Predictors

### 3.1. Clinical Predictors

Clinical features of stroke subjects that predict late epileptic seizures are considered the most significant. Graham et al. [12] performed an analysis of over 3000 stroke patients from the South London Stroke Register, following them for up to 12 years. The incidence of poststroke epilepsy was estimated at 6.4%. The authors showed that age of <65 years was significantly associated with a higher risk of epilepsy related to stroke compared to subjects over 85 years old. Another important predictor was stroke subtype. The cumulative risk of poststroke epilepsy was significantly increased in subjects with total anterior circulation infarct (TACI) followed by subarachnoidal hemorrhage and intracranial hemorrhage. The other subtypes of ischemic stroke were characterized by a lower cumulative risk of seizures. The authors also concluded that some clinical symptoms related to stroke, such as dysphasia, neglect, and visual field defect, which may indicate an involvement of the cerebral cortex, were associated with a higher incidence of poststroke epilepsy. Another clinical feature related to higher risk of seizures was stroke severity, as measured by the Glasgow Coma Scale (GCS) and Barthel Index. Lower scores on the GCS and Barthel Index, indicating severe disability and poor clinical condition, were predictors of epilepsy.

Conrad et al. [13], in a cohort study of almost 600 stroke subjects followed for up to 30 months, estimated a total prevalence of epilepsy related to stroke of 11.6%. They showed that subjects who developed seizures were significantly younger compared to the non-seizure subgroup. A severe initial clinical condition measured by the National Institute of Health Stroke Scale (NIHSS) was an independent predictor of late epileptic seizures. The authors also found that thrombotic (macroangiopathic) etiology of stroke is significantly associated with higher incidence of seizures compared to microangiopathic or cardioembolic etiology of stroke. Intracranial hemorrhage was significantly more common in the seizure group than in the non-seizure group, whereas ischemic stroke did not differ significantly in incidence in the groups. Different analyzed subtypes of ischemic strokes (anterior or posterior territory, middle cerebral artery territory, right or left hemisphere) had similar incidence in seizure and non-seizure groups. The authors also reported no relationship between higher incidence of seizures and pre-existing vascular risk factors for cerebrovascular disease (such as hypertension, diabetes, atrial fibrillation, smoking, and obesity).

Kammersgaard et al. [14] investigated 1195 stroke subjects followed for up to seven years in a prospective study and estimated poststroke epilepsy at only 3.2%. The authors reported that, in a univariate analysis, higher incidence of epileptic seizures was associated with younger age and cerebral hemorrhage. In a multivariate regression analysis, they reported that independent predictors of stroke-related epilepsy are younger age, initial severe clinical status (measured by Scandinavian Stroke Scale), cerebral hemorrhage, and incidence of early seizures after stroke. Surprisingly, ischemic heart disease and atrial fibrillation, indicating a cardioembolic background of stroke, were significantly less common in the seizure group.

Bladin et al. [11], in a prospective, multicenter, cohort study, included 1897 stroke subjects followed for up to nine months. Cardioembolic stroke and younger age were not associated with an increased rate of epileptic seizures. Subjects with hemorrhagic stroke were more likely to develop seizures than ischemic stroke subjects. Initial stroke disability, measured by the Canadian Neurological Scale, was an independent predictor of late seizures. Long-term unfavorable functional outcome measured by modified Rankin Scale (mRS) was significantly more common in the group with seizures.

De Reuck et al. [15], in a retrospective, observational study, included 476 subjects with stroke. The authors did not report the impact of stroke etiology, age, or well-known vascular risk factors (including cardioembolism) on the seizure incidence, which was overall estimated at 23%. They revealed that a subtype of ischemic stroke, partial anterior circulation syndrome, was an independent factor associated with an increased rate of poststroke epilepsy. In contrast to the above reports, they showed no impact of hemorrhagic stroke or total anterior circulation syndrome on a higher prevalence of stroke-related seizures. The stroke severity on admission, as assessed by NIHSS and stroke disability (mRS) on discharge, did not differ significantly between groups with or without seizures.

Cheung et al. [16], in a retrospective analysis of 1000 stroke subjects from the acute stroke registry, reported that male sex, older age, partial anterior and total anterior circulation infarctions were associated with the development of late seizures. In a multivariate analysis, only sex (male) was an independent predictor of stroke-related seizures. Nevertheless, no increased risk of epilepsy was found in subjects with hemorrhagic stroke.

Roivainen et al. [17] analyzed the data from the Helsinki Young Stroke Registry and reported that partial anterior and total anterior circulation infarctions, hemorrhagic infarct, sex (male), hyperglycemia, and history of early seizures after the onset of stroke were independent risk factors for the development of poststroke epilepsy. The etiology of ischemic stroke and well-known risk factors for cerebrovascular diseases were not related to epileptic seizures. Moderate and severe strokes (assessed by NIHSS) were more common in the group with seizures, but in a Cox regression stroke severity did not significantly affect the incidence of stroke-related epilepsy.

Wang et al. [18], in a large, multicenter study, among 2474 Chinese stroke subjects, did not show any significant relationships of poststroke seizures with clinical features. They reported that patients who developed seizures were significantly older than nonseizure subjects and had a similar clinical condition assessed by NIHSS. Likewise, stroke type, including hemorrhagic stroke, did not affect the incidence of late seizures.

Serafini et al. [19] revealed that younger age and cortical location of ischemic stroke were significantly related to poststroke epilepsy in a multivariable analysis. Early seizures were significant predictors of the development of stroke-related epilepsy, but only among hemorrhagic stroke subjects.

Jungenhulsing et al. [20], in a prospective analysis of the Erlange Stroke Project registry, reported that only stroke severity (assessed 5–7 days after onset based on the Barthel Index) is an independent predictor of stroke-related epilepsy. No significant impact of age, sex, stroke type, or comorbidities was found.

Lossius et al. [21], in a prospective study focused on 484 stroke subjects followed for up to eight years, showed that only stroke severity on admission (measured by the Scandinavian Stroke Scale) was an independent factor that led to a he higher risk of poststroke epilepsy. No relationships with age, sex, or risk factors for vascular diseases was reported.

### 3.2. Radiological Predictors

Conrad et al. [13] showed that neuroimaging, cortical involvement, and anterior or posterior localization of stroke did not differ significantly between seizure and nonseizure groups. The impact of a size of the ischemic focus on the prevalence of epileptic seizures was not analyzed.

Kammersgaard et al. [14] showed that a large lesion (increase by >1 cm in diameter) in neuroimaging is an independent predictor of epileptic seizures. However, cortical involvement was not significantly associated with poststroke epilepsy in uni- or multivariate analyses. Similarly, Lossius et al. [21] found no significant impact of the cortical location of lesions on the risk of stroke-related epilepsy.

In contrast, Bladin et al. [11] showed that cortical involvement was significantly more common in the seizure group. In a multivariate analysis, cortical location became an independent predictor of stroke-related seizures. Moreover, patients with a larger size of ischemic focus in computed tomography (CT) exhibited a higher percentage of late-onset seizures.

Chen et al. [22] investigated the possibility of using the Alberta Stroke Program Early CT Score (ASPECTS) to assess the extent of acute cerebral ischemia in predicting late seizures. The authors revealed that ASPECTS, both on admission and 24 h after the onset, are associated with a higher incidence of stroke-related seizures. Moreover, in a multivariate regression, the extent of cerebral ischemia, as measured by ASPECTS, was the only independent predictor of the development of poststroke epilepsy. Additionally, a significant correlation was found between cortical involvement at 24 h after the onset of stroke and the occurrence of seizures. However, no significant relationship between cortical involvement on admission and the development of seizures was reported.

Wang et al. [18] analyzed the impact of radiological features on the occurrence of poststroke seizures. They found that cortical involvement and large lesion size (>3.5 cm in diameter) were independent predictors of epileptic seizures.

Cheung et al. [16] showed that cortical location and large lesion size were related to poststroke epilepsy. However, only cortical involvement was an independent radiological predictor in a multivariate analysis.

Okuda et al. [23], in a retrospective study of 448 stroke subjects, noted that a cortical localization of ischemic stroke and large infarcts involving the middle cerebral artery were significantly associated with the development of late epileptic seizures.

A summary of the presented findings is shown in Table 1.

### 3.3. Prognostic Models Based on Combinations of the Above Predictors

Some authors decided to collect several epilepsy-related variables to develop a prognostic model that would be a combination of clinical and radiological features. The first prognostic model was proposed in 2010 by Strzelczyk et al. [24], who analyzed a cohort of 264 subjects with both ischemic and hemorrhagic stroke and followed them for up to one year. The Poststroke Epilepsy Risk Scale (PoSERS) includes seven items: supratentorial stroke, cortical hemorrhage, late seizure presence, cortical or subcortical ischemic stroke, ischemia with ongoing neurological deficit, stroke-related neurological deficit mRS of >3, and early seizure presence. However, the first three items were double-weighted in an analysis of predictive value. PoSERS was related with 70% sensitivity and 99.6% specificity. The advantage of this scale is its attempt to combine ischemic and hemorrhagic stroke as well as clinical and radiological features. Among the limitations are the small group of respondents, short follow-up period, lack of validation, and taking into account only recurrent seizures, which is not in accordance with the updated definition of epilepsy.

In 2014, Haapaniemi et al. [25] presented the CAVE scale, which is useful for the prediction of poststroke epilepsy among hemorrhagic stroke subjects. The CAVE scale was developed based on 993 cohort subjects followed for up to 2.7 years. It included four variables: cortical involvement, age of <65 years, hemorrhagic focus volume of >10 mm at baseline, and early seizure presence within first seven days of intracranial hemorrhage. Each item is scored from 0 to 1 point and has an equal value. The maximum score is 4, and a score of 2 or more is related to 81% sensitivity and 89% specificity. The risk of poststroke epilepsy was estimated at 0.6%, 3.6%, 9.8%, 34.8%, and 46.2%, corresponding to CAVE scores 0 to 4, respectively. The CAVE score seems to be an easy tool to use in clinical practice, consisting of obvious variables, and it has been validated. However, it only applies to hemorrhagic strokes, which limits its use. In 2020, Kwon et al. [26] analyzed data from 2507 subjects from the Ethnic/Racial Variations of Intracerebral Hemorrhage (ERICH) prospective study to validate the prognostic value of the CAVE score. They found that the CAVE score was significantly associated with the development of late seizures (odds ratio (OR) = 2.5, 95% confidence interval (CI) 2.0–3.2; *p* < 0.0001). Additionally, the authors suggested replacing early seizures with surgical evacuation of the hematoma, which more significantly affects the risk of late seizures. A newly created diagnostic tool, the CAVS scale, allowed us to achieve an even higher predictive value (OR = 2.8, 95% CI 2.2–3.5).

In 2018, Galovic et al. [27] presented the SeLECT score based on 1200 ischemic stroke subjects followed for up to five years. The prognostic model included five parameters: severity of stroke assessed by NIHSS on admission (scored 0–2 points), large-artery atherosclerotic etiology (scored 0–1 points), early seizures (scored 0–3 points), cortical involvement (scored 0–1 points), and middle cerebral artery involvement (scored 0–1 points). The maximum score was nine points and was associated with 63% risk of late epileptic seizures in the first year after the onset of stroke and 83% cumulative risk within five years. The authors deliberately excluded parameters that, despite having a significant predictive effect, have no practical clinical application, such as the size of a focus. They also excluded parameters that do not support sufficient validation data. The most relevant variables related to higher risk of poststroke epilepsy from previous reports were taken into account, including age, sex, and thrombolysis treatment. According to the multivariable regression analysis, five items were selected. The SeLECT score was successfully validated (with a concordance statistic of 0.77) and achieved good calibration and discrimination values. The SeLECT score is currently considered to be the most developed prognostic model of poststroke epilepsy. However, it only affects ischemic stroke and at least two items, large-vessel etiology and the territory of the middle cerebral artery, may raise doubts, as they are not often described in other studies on this topic as significant risk factors and mentions of their role are sporadic [13,18,28].

### 3.4. Meta-Analysis Related to Clinical and Radiological Predictors

Besides observational and cohort studies, we also found two large meta-analyses that summarized previous findings on both clinical and radiological predictors of poststroke epilepsy. A meta-analysis made by Ferlazzo et al. [29] showed that hemorrhagic stroke was associated with an almost doubled risk of epileptic seizures and early seizures were associated with a fourfold increased risk of poststroke epilepsy. The authors reported cortical involvement as an independent factor that causes an almost fourfold increase in the risk of stroke-related epilepsy. In another meta-analysis, Zhang et al. [30] found that only stroke severity on admission was an independent predictor of poststroke epilepsy, while the other factors did not lead to higher risk of epileptic seizures. The probability of epileptic seizures was similar regardless of stroke type, sex, and well-known risk factors for vascular diseases. The authors revealed that cortical involvement is an independent predictor of stroke-related epileptic seizures, while similar relationships were not noted in the case of the size of ischemic focus.

### 3.5. Electroencephalographic Predictors

Electroencephalography (EEG) still remains the basic diagnostic test performed in subjects with epilepsy. Disturbances in EEG curves are not only diagnostic, but also prognostic and sometimes decisive for the initiation of antiepileptic treatment. For many years, research has been carried out to find characteristic abnormalities in the EEG recording, occurring in the early stages of stroke, which could be used to predict the risk of poststroke epilepsy. Bentes et al. [31] included 151 stroke subjects in a prospective study and performed video EEG within the first 72 h after the onset. They identified EEG background activity asymmetry and interictal epileptiform activity as two independent predictors of stroke-related epilepsy, increasing the risk of occurrence by more than threefold. However, the authors followed the subjects for only up to one year. De Reuck et al. [32], in a retrospective analysis of 385 stroke subjects, followed for up to three years, revealed that frontal intermittent rhythmic delta activities (FIRDA) and diffuse slowing of background EEG activity are significantly related to a higher risk of developing late epileptic seizures. It is also worth noting that, compared to the group without seizures, patients who experienced a late seizure significantly less frequently had a normal EEG pattern (5.1% vs. 53.8%; *p* < 0.001). Other abnormalities, such as periodic lateralized epileptiform discharges (PLED) or focal background slowing, did not affect the risk of stroke-related epilepsy. In contrast, Onder et al. [33] and Strzelczyk et al. [24] found no impact of EEG abnormalities in the acute stage of stroke on the occurrence of late epileptic seizures and concluded that the predictive utility of this method was questionable.

### 3.6. Genetic Predictors

It seems that genetic predisposition may contribute to an increased risk of poststroke epilepsy. There are reports showing the influence of family history on the incidence of late epileptic seizures [34]. Several authors have succeeded in identifying polymorphic genetic variants that may significantly predict stroke-related seizures. Yang et al. [35] analyzed mitochondrial aldehyde dehydrogenase 2 (ALDH2) rs671 polymorphism in terms of the risk of poststroke epilepsy. They revealed that rs671 A allele carriers were associated with a higher risk of late epileptic seizures, as a result of higher levels of plasma 4-hydroxynonenal, an ALDH2 substrate, involved in oxidative stress reactions and cerebral ischemia. Zhang et al. [36] reported the significant role of the T allele of the CD40-1 C/T polymorphism in predicting late seizures as a result of the higher expression of CD40 mRNA, involved in prothrombotic conditions and oxidative stress.

### 3.7. Laboratory Predictors

Blood biomarkers related to poststroke epilepsy were not as widely investigated as in early seizures. Abraira et al. [37], in a cohort study with a total of 895 stroke subjects, evaluated a panel of 14 blood biomarkers of neuroinflammation processes that happen in the brain after stroke. They reported that high plasma levels of endostatin and low levels of heat shock 70 kDa protein (Hsc70) and S100 calcium-binding protein B (S100B) were significantly associated with the development of stroke-related epileptic seizures. Zhang et al. [36] found enhanced plasma levels of soluble CD40 ligand (sCD40L), linked to inflammation processes and vascular dysfunction, in the group with late epileptic seizures compared to stroke subjects without seizures.

### 3.8. Predictors Related to Treatment

Specific treatment performed in the acute phase of a stroke may affect the development of poststroke epilepsy. Keller et al. [38] showed that late epileptic seizures occurred significantly frequently in stroke subjects who underwent intravenous thrombolytic treatment in the acute period of cerebral ischemia compared to stroke subjects without reperfusion therapy. However, in a multivariate analysis, this finding was lost. Similarly, Naylor et al. [39], in a retrospective, two-center study, investigated the role of specific stroke treatment in predicting poststroke epilepsy. In a multivariable analysis, they reported that subjects who underwent reperfusion therapy were more likely to develop late seizures compared to standard care stroke subjects. In particular, intravenous thrombolysis was associated with a threefold higher risk of stroke-related seizures. In contrast, Bentes et al. [40] and Tan et al. [41] did not reveal any correlation between thrombolysis in the acute stage of stroke and a higher risk of late seizures. Moreover, Nesselroth et al. [42] showed that thrombolysis treatment significantly decreased the risk of poststroke epilepsy.

## 4. Discussion

In view of the conflicting current reports on multifactorial predictors of poststroke epilepsy, it seems that the development of a common and unified prognostic model to stratify subjects in the acute phase of a stroke is a very difficult, if not impossible, task. Although most research concerned clinical and radiological predictors, the significance was not confirmed in any of the analyzed studies. In our opinion, this results from the high heterogeneity of the studies, including the diverse study populations, variable sample size, different assessment methods, variable follow-up periods, and different types of research. Therefore, the development and validation of prognostic instruments, in particular CAVE and SeLECT, should be appreciated and respected. Despite some drawbacks, in light of the presented data, they contain the most important highlighted and tested predictors. In addition, they are calculated from commonly available and routinely assessed variables, which is their great advantage. In the future we should strive to carry out as many multicenter studies as possible, the purpose of which will be to objectively validate these scales and enable them to be used as widely as possible in everyday medical practice. Only such a procedure will make it possible to modify and refine existing forecasting models. It seems that the further extension of the existing scales with electroencephalographic, genetic, or laboratory factors is unjustified. Furthermore, the few, often contradictory, reports on the role of EEG abnormalities—especially laboratory and genetic factors—do not support including them in prognostic models. Despite reports on the increased risk of poststroke epilepsy, thrombolytic treatment will remain the gold standard of acute stroke therapy, bringing about more benefits. Nevertheless, it is worth considering supplementing these scales with more sophisticated and advanced radiological methods in the future due to their significant prognostic importance. Moreover, an alternative quantitative EEG analysis based on computerized algorithms, such as fast Fourier transformation, may soon replace the standard qualitative EEG and achieve a higher predictive value. The question remains whether existing models should be developed separately for ischemic and hemorrhagic stroke or whether one should strive to unify them. However, taking into account the extremely different pathophysiological mechanisms of both diseases, it seems that their separation should be maintained.

Stratification of the risk of developing late epileptic seizures after stroke based on improved and validated existing prognostic models would enable the identification of select stroke subjects who may particularly benefit from antiepileptic treatment. This could help with developing a more individualized approach to these patients and improve the quality of care based on standardized methods. The prognostic model may also be useful in optimizing selection for prospective clinical trials and studies regarding epileptogenesis among stroke subjects.

## 5. Conclusions

Determination of the risk of late epileptic seizures after stroke is crucial because of its harmful nature. At the moment, it seems impossible to develop an objective, reliable, and unified prognostic model that covers all the relevant variables that may influence the development of poststroke epilepsy, due to the conflicting nature of the available data on the multifactorial predictors of late epileptic seizures. Further research is thus needed to confirm the predictive value of existing prognostic models of stroke-related epilepsy and seek to refine them for widespread use in stroke units.

## Figures and Tables

**Figure 1 ijerph-18-01079-f001:**
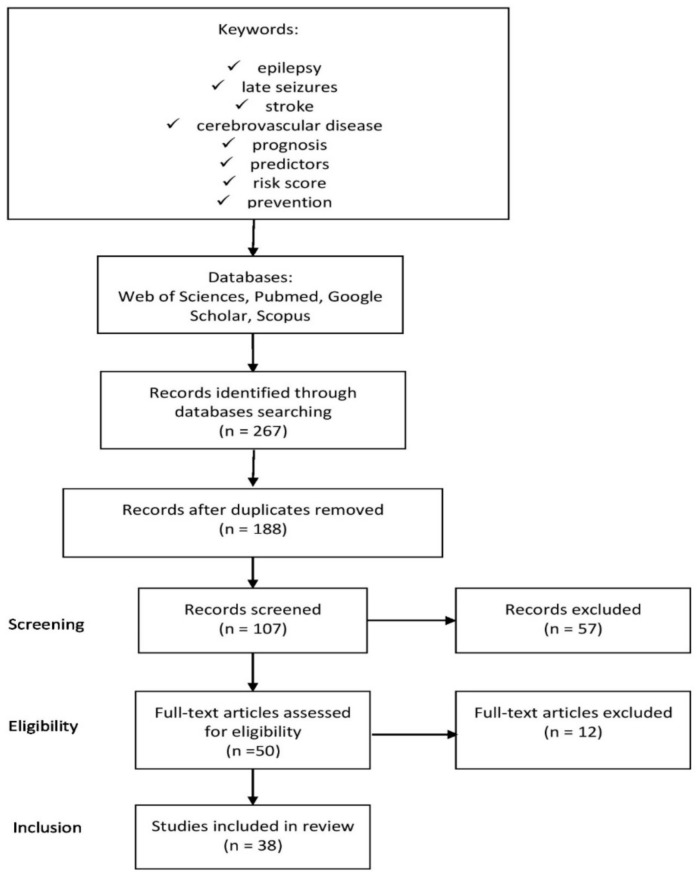
Diagram showing the process of selecting the appropriate literature.

**Table 1 ijerph-18-01079-t001:** Summary of the most relevant papers regarding clinical and radiological predictors of poststroke epilepsy.

Article (Year) [Reference]	Type	Sample Size	Follow-Up	Clinical Predictors	Radiological Findings (Predictors)
Graham et al. (2013) [12]	prospective	3310	12 years	young age of <65 years TACI, subarachnoidal hemorrhage, intracranial hemorrhage, lower scores on Glasgow Coma Scale and Barthel Index	n/a
Conrad et al. (2013) [13]	cohort	593	3.5 years	young age, severe initial clinical condition estimated by NIHSS, thrombotic (macroangiopathic) etiology of stroke, intracranial hemorrhage	no significant impact of multiple changes in neuroimaging, cortical involvement, anterior or posterior localization of stroke
Kammersgaard et al. (2005) [14]	prospective, observational	1195	7 years	younger age, initial severe clinical status (measured by Scandinavian Stroke Scale), cerebral hemorrhage and incidence of early seizures after stroke	large lesion (increase by >1 cm in diameter)
Bladin et al. (2000) [11]	prospective, cohort	1897	9 months	hemorrhagic stroke, initial stroke disability measured by Canadian Neurological Scale, long-term unfavorable functional outcome measured by modified Rankin Scale (mRS)	cortical location and larger size of ischemic focus in computed tomography
De Reuck et al. (2005) [15]	retrospective, observational	476	4 years	PACI stroke	n/a
Cheung et al. (2003) [16]	retrospective	1000	12 months	male sex, older age, partial anterior and total anterior circulation infarctions	cortical location and large lesion size
Roivainen et al. (2013) [17]	observational, cohort	978	10 years	partial anterior and total anterior circulation infarctions, hemorrhagic infarct, male sex, hyperglycemia and history of early seizures after the onset of stroke,	n/a
Wang et al. (2013) [18]	retrospective, multicenter	2474	2 years	older age among stroke subjects	cortical involvement and large lesion size (>3.5 cm in diameter)
Serafini et al. (2015) [19]	prospective	782	2 years	younger age, early epileptic seizures (only among hemorrhagic stroke)	cortical involvement
Jungenhulsing et al. (2013) [20]	prospective	1815	2 years	stroke severity (assessed on 5–7 days after the stroke onset on Barthel index)	n/a
Lossius et al. (2005) [21]	prospective	484	8 years	stroke severity on admission (measured by Scandinavian Stroke Scale)	no significant impact of cortical involvement
Chen et al. (2017) [22]	prospective, cohort	348	3 years	n/a	the extent of cerebral ischemia, measured by ASPECTS, cortical involvement at 24 h after the onset of stroke
Okuda et al. (2012) [23]	retrospective	448	18 months	n/a	cortical localization of ischemic stroke and large infarcts involving middle cerebral artery

PACI—partial anterior circulation infarct; TACI—total anterior circulation infarct, NIHSS—the National Institute of Health Stroke Scale, mRS—modified Rankin Scale, ASPECTS—Alberta Stroke Program Early Computed Tomography Score, n/a—not applicable.

## Data Availability

Not applicable.

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
