# Peer review of "Multifactorial Predictors of Late Epileptic Seizures Related to Stroke: Evaluation of the Current Possibilities of Stratification Based on Existing Prognostic Models—A Comprehensive Review"

_ijerph, 2021, doi:10.3390/ijerph18031079_

Round 1

Reviewer 1 Report

The review entitled: “Multifactorial predictors of epilepsy related to cerebrovascular diseases - a current attempt to develop a common and unified prognostic model” is well-written in good mannered way. However, some issues need to be addressed before final publication.

The authors highlight the difficulties for finding standardization in predictive models, those allowing identification of patients in the acute phase of stroke at high risk for developing late seizures.

Figure 1 needs clarification, please use clear flow diagram; it should be re-fine and explain clearly in the legend. There are also some typographical errors that should be corrected before the review is submitted for publication.

Reviewer 2 Report

In the present paper Wisniewski and Jatuzis review the recent efforts of finding adequate predictors for epilepsy seizures onset after stroke.

The review is comprehensive and well referenced, however it not of easy interpretation due to the several literature presented throughout the paper. The authors are encouraged to insert a table for each section of the results to summarize the cited literature for a fast and synoptic vision of the topic.

When analysing the single predictors the authors should leave out the meta analysis on the topic, and maybe have a separate paragraph to analyze them. There are high chances that the meta-analyses elaborate and discuss the data reviewed in the paragraph.

Finally, the title is misleading: the current title suggests an effort to propose an original prognostic model and the term cerebrovascular disease is not exclusive for stroke, which in turn is the main focus of the paper. The authors should reword the title accordingly to the paper contents.

Round 2

Reviewer 2 Report

No further comments

Author Response

As the Reviewer positively considered our revised version of the manuscript and has "no further comments"  I would like to thank Him for  valuable comments that significantly contributed to the improvement of our paper.